# Application of a Lipopolysaccharide (LPS)-Stimulated Mitogenesis Assay in Smallmouth Bass (*Micropterus dolomieu*) to Augment Wild Fish Health Studies

Cheyenne R. Smith [1],*, Christopher A. Ottinger [2], Heather L. Walsh [2], Patricia M. Mazik [3] and Vicki S. Blazer [2]

1    Division of Forestry and Natural Resources, West Virginia University, 333 Evansdale Drive, Morgantown, WV 26506, USA
2    U.S. Geological Survey, Eastern Ecological Science Center at Leetown Research Laboratory, 11649 Leetown Road, Kearneysville, WV 25430, USA
3    U.S. Geological Survey, West Virginia Cooperative Fish and Wildlife Research Unit, West Virginia University, Morgantown, WV 26506, USA
*    Correspondence: crsimpson@usgs.gov

**Abstract:** The utility of a functional immune assay for smallmouth bass (*Micropterus dolomieu*) lymphocyte mitogenesis was evaluated. Wild populations in the Potomac River have faced disease and mortality with immunosuppression from exposure to chemical contaminants a suspected component. However, a validated set of immune parameters to screen for immunosuppression in wild fish populations is not available. Prior to use in ecotoxicology studies, ancillary factors influencing the mitogenic response need to be understood. The assay was field-tested with fish collected from three sites in West Virginia as part of health assessments occurring in spring (pre-spawn; April–May) and fall (recrudescence; October–November). Anterior kidney leukocytes were exposed to lipopolysaccharide (LPS) from *E.coli* O111:B4 or mitogen-free media and proliferation was measured using imaging flow cytometry with advanced machine learning to distinguish lymphocytes. An anti-smallmouth bass IgM monoclonal antibody was used to identify IgM+ lymphocytes. Lymphocyte mitogenesis, or proliferative responses, varied by site and season and positively and negatively correlated with factors such as sex, age, tissue parasites, and macrophage aggregates. Background proliferation of IgM− lymphocytes was negatively correlated to LPS-induced proliferation in both seasons at all sites, but only in spring for IgM+ lymphocytes. The results demonstrate that many factors, in addition to chemical contaminants, may influence lymphocyte proliferation.

**Keywords:** fish; immune responses; mitogenesis; imaging flow cytometry

**Key Contribution:** The manuscript describes a functional immune assay optimized and validated for wild smallmouth bass (*Micropterus dolomieu*). The assay can be integrated into fish health assessments to investigate immunosuppression. The study demonstrated the importance of evaluating variables such as sex, age, and parasite load, as well as season and site when utilizing immune indicators in ecotoxicological studies.

## 1. Introduction

Globally, there is an increasing trend in the observations of infectious diseases, particularly in wild populations [1], including smallmouth bass (*Micropterus dolomieu*) [2–5]. Microorganisms, viruses, and parasites are common causes; however, disease outbreaks in fish can also be exacerbated by environmental conditions, including temperature changes and chemical contaminants [6,7]. Environmental factors may directly influence the host immunocompetence making them more susceptible to pathogens, but also indirectly affect disease resistance by impacting pathogen presence, concentration, and/or virulence [8,9].

In addition, environmental factors may act on parasitic diseases directly or indirectly by affecting intermediate hosts and transmission vectors [10]. When so many factors are involved, it can be hard to determine underlying risk factors for disease or mortality in wild fish populations, and validated methods for determining immunomodulation in the context of many stressors are not established.

There is increasing interest in including fish immune responses in ecotoxicological studies; however, historically, fish immunotoxicity studies have been conducted in the laboratory and have focused on the innate immune system [11,12]. To reliably utilize immune function assays in wild populations, it is necessary to understand the role of factors other than chemical exposure in modulating responses. Mitogenesis is an indicator of adaptive immunity and one advantage to the optimized method is the ability to distinguish responses of immunoglobulin M positive (IgM+) lymphocytes (members of the B-cell lineage that express IgM on their plasma membrane) from immunoglobulin M negative (IgM−) lymphocytes (cells of the B-cell lineage that do not express surface IgM or members of other lymphocyte lineages such as T-cells and natural killer cells). This distinction is important because commercial markers to label leukocyte types are not readily available for most wild fish species including smallmouth bass.

Imaging flow cytometry with advanced machine learning was also used in this study to ensure lymphocyte proliferation was being measured. Imaging flow cytometry combines attributes of standard flow cytometry (statistical power, fluorescent intensity, and speed) with the abilities of fluorescence microscopy (detailed imagery, spatial distribution of fluorescent signals, and morphological features) to rapidly analyze and generate a collection of multicolor images for individual cells in suspension [13–16]. Multiple applications of imaging flow cytometry have been established for various species; however, methods for investigating piscine leukocytes with imaging flow cytometry are limited to studies primarily addressing phagocytosis [17–21] and our previous studies addressing mitogenesis (without distinguishing lymphocyte populations) and respiratory burst [22,23].

This study describes and tests a functional mitogenesis assay using 5-ethyl-2′-deoxyuridine (EdU) to detect and measure adaptive immunity in wild smallmouth bass (*Micropterus dolomieu*), an economically important sportfish. The anterior kidney was collected from fish at three field sites where fish health assessments were being conducted. We evaluated associations of lymphocyte mitogenesis with the season, sex, age, tissue parasite, and macrophage aggregate density to better understand variables that may influence responses in immunotoxicological studies.

## 2. Materials and Methods

### 2.1. Field Sampling

Wild fish sampling took place in conjunction with ongoing fish health assessment and monitoring studies at three sites in West Virginia (Figure 1): Cheat River (CH) at Hannahsville, WV (39.24525, −79.70781), South Branch of the Potomac River at Petersburg, WV (SB1) (39.00025, −79.08666) and South Branch of the Potomac River at Moorefield, WV (SB3) (39.10367, −78.95891). The sites were selected based on proportions of agricultural, developed, and forested land use and previous health status of smallmouth bass. Smallmouth bass populations in the South Branch of the Potomac Rivers have undergone large-scale episodic mortality and disease events of adults since 2002 [2,24]. The SB3 site has been part of a larger long-term USGS monitoring study to investigate smallmouth bass health issues in the Chesapeake Bay watershed since 2013. CH was selected as an out-of-basin site with less surrounding agricultural land use. Fish kills or major health problems have not been observed at the CH site; however, *Aphanomyces invadans*, the cause of epizootic ulcerative syndrome, was identified in two individuals in 2020 [5]. A lower incidence of estrogenic endocrine disruption or testicular oocytes (intersex) was also observed in the Cheat when compared to the South Branch Potomac [25].

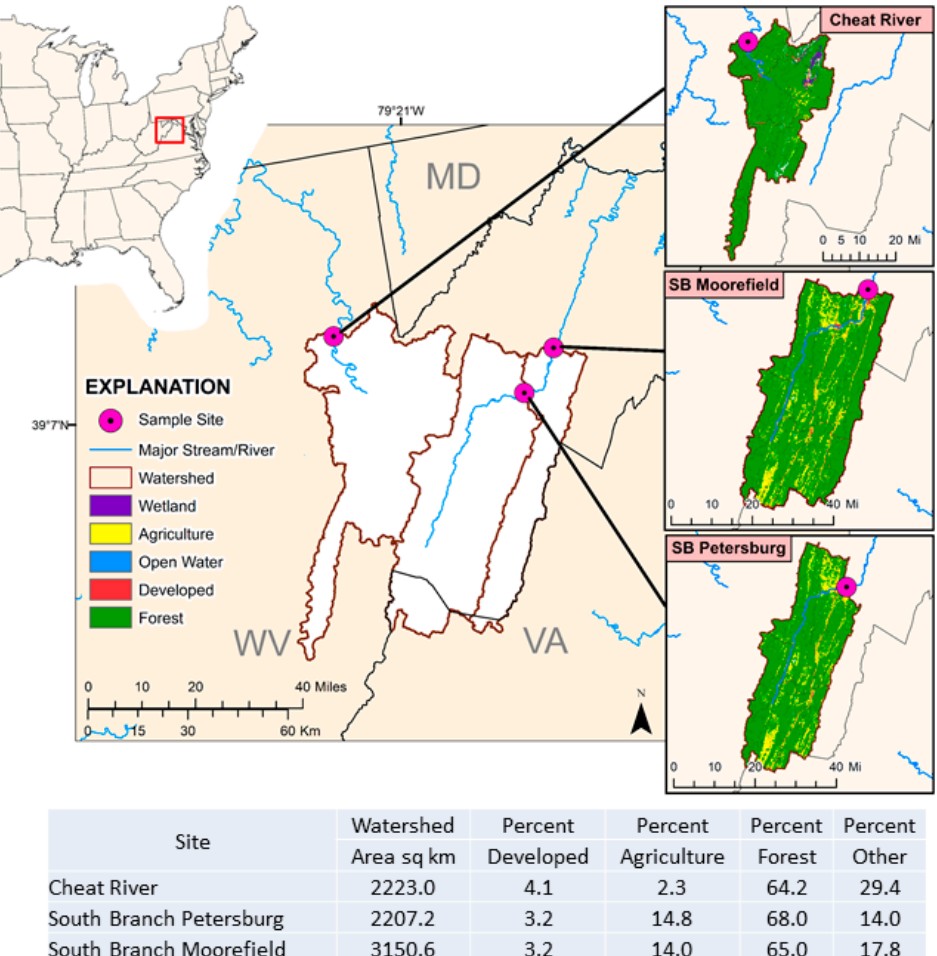

| Site | Watershed Area sq km | Percent Developed | Percent Agriculture | Percent Forest | Percent Other |
|---|---|---|---|---|---|
| Cheat River | 2223.0 | 4.1 | 2.3 | 64.2 | 29.4 |
| South Branch Petersburg | 2207.2 | 3.2 | 14.8 | 68.0 | 14.0 |
| South Branch Moorefield | 3150.6 | 3.2 | 14.0 | 65.0 | 17.8 |

**Figure 1.** Fish collection sites at Cheat River near Hannahsville, West Virginia (CH); South Branch Potomac near Petersburg, West Virginia (SB1); and South Branch Potomac near Moorefield, West Virginia (SB3) with major land cover in the upstream catchments of study sites.

Land cover data were downloaded from the 2019 National Land Cover Database at https://www.mrlc.gov/data (accessed on 10 October 2022) and upstream catchments were generated by manually selecting National Hydrography Dataset Plus (NHD+) Catchments upstream of each site along NHD flowlines at https://www.horizon-systems.com/NHDPlus/NHDPlusV2_data.php (accessed on 10 October 2022). Summaries of land cover were generated using a zonal histogram tool (ArcMap; version 10.6) and are presented as percent. Land cover in the upstream catchments of the three sites were primarily forested with low-developed land use; however, the South Branch sites (SB1 and SB3) had more agricultural lands than the Cheat River (CH) site (Figure 1).

Attempts were made to collect 20 adult smallmouth bass from each of the three sites by boat electrofishing in spring (April–May) before spawning and fall (October–November) during recrudescence of 2019. Captured fish were immediately placed in a live well and transported to shore to be necropsied on site following a protocol by Blazer, et al. [26] and explained more in-depth by Smith, et al. [23]. In short, fish were euthanized, measured total length to the nearest mm, weighed to the nearest gm, bled from the caudal vessels, and any gross abnormalities were recorded. Anterior kidney tissue was aseptically removed in the field, homogenized into a single cell suspension, transported to the lab on ice, and maintained at 4 °C overnight for leukocyte isolations the next day. In the field following removal of anterior kidney tissue, remaining tissues which included gills, gonads, liver, spleen, posterior kidney, and any visibly abnormal tissues were removed for histology and

preserved (in Z-fix™; Anatech LTD, Battle Creek, MI, USA). Gonads were examined to confirm sex. Lapilli otoliths were collected for age analyses [26].

### 2.2. Laboratory Analyses

### 2.2.1. Estimating Disease Indicators with Histology

Tissues fixed for histological analyses were routinely processed, embedded into paraffin, sectioned at 5 μm, and stained with hematoxylin and eosin [27]. Parasites in liver, spleen, and posterior kidney and splenic macrophage aggregates (MA) were counted. Parasites were counted in 3 fields of view at 4× and the number per sq mm was calculated. MA was counted in 5 fields of view at 10× and the number per sq mm was calculated. Parasite and MA density were calculated using the area of the field of view (FOV) at each magnification. Parasites or MA not fully in the FOV were not counted.

### 2.2.2. Leukocyte Isolations

Anterior kidney leukocytes were isolated following a procedure modified from Sharp, et al. [28]. Isolated leukocytes were utilized for a full suite of immune functional assays including two for innate immunity [23] in addition to the mitogenesis assay to investigate multiple aspects of the immune response at the same time. Briefly, homogenized anterior kidney samples were washed three times and then layered on top of a 32 percent density gradient (Percoll®; Sigma-Aldrich, St. Louis, MO, USA) concentration in Hank's Balanced Salt Solution (MP Biomedicals, Irvine, CA, USA) for leukocyte separation. A density gradient specifically for lymphocyte separation was not chosen as other leukocytes (i.e., granulocytes) were needed for innate immune function assays being performed on the same fish detailed in Smith, et al. [23]. All functional immune assays were performed with the same set of isolated cells. Isolated leukocytes were counted, total number of viable leukocytes were determined using an automated cell counter (Countess™ II; ThermoFisher, Waltham, MA, USA), and leukocytes were resuspended at $2 \times 10^7$ cells mL$^{-1}$ in culture medium (Leibovitz's L-15 medium containing 290 μg mL$^{-1}$ L-glutamine, 100 U ml$^{-1}$ penicillin, 100 μg mL$^{-1}$ streptomycin, and 5 percent fetal bovine serum, FBS; L-15/5% P/S).

### 2.2.3. Mitogenesis

Mitogenesis was evaluated following an EdU-based assay protocol (Click-iT™ Plus EdU Alexa Fluor® 647 Flow Cytometry Assay Kit, Molecular Probes, Eugene, OR, USA) optimized for smallmouth bass [23] with modifications to require fewer isolated leukocytes per treatment, strengthen/increase fluorochrome signal, and label cells with immunoglobulin for distinguishing responses of IgM+ and IgM− lymphocytes. The final panel developed for the mitogenesis assay using smallmouth bass anterior kidney leukocytes included several fluorochromes to label and detect 1) IgM+ cells (Alexa Fluor® 488; AF488), 2) G$_2$/mitotic phase (G2/M; two sets of paired chromosomes per cell prior to cell division) cells (FxCycle™ propidium iodide with RNase; PI/RNase; ThermoFisher, Waltham, MA, USA), and 3) EdU+ cells (Alexa Fluor® 647; AF647).

Isolated leukocytes in culture medium (L-15/5% P/S) were plated at 25 μL well$^{-1}$ ($5 \times 10^5$ cells well$^{-1}$) in Falcon 384-well tissue culture plates (Corning Cat. No. 353961; ThermoFisher, Waltham, MA). This format required fewer cells ($5 \times 10^5$ cells well$^{-1}$) per well and allowed more assays to be performed with the total number of isolated cells relative to the 96-well format ($1 \times 10^6$ cells well$^{-1}$) reported previously [23]. Isolated leukocytes were added to tissue culture plates in duplicate for each treatment (mitogen and negative controls) and pooled prior to analysis with imaging flow cytometry. A minimum of 0.2 mL of isolated leukocytes resuspended at $2 \times 10^7$ cells ml$^{-1}$ was required to complete the mitogenesis assay in the 384-well plate format. In some cases when leukocyte yields were low, mitogenesis was not assessed.

Once plated, leukocytes were treated with either 25 μL well$^{-1}$ of mitogen or mitogen-free media (negative control wells). Plates were protected from light for the remainder of the assay. Treatments included lipopolysaccharide from *Escherichia coli* O111:B4 (LPS)

at 100 μg mL$^{-1}$ (2.5 μg well$^{-1}$ final concentration) or mitogen-free culture medium as negative control (L-15/5%). LPS, a bacterial endotoxin, was chosen to mimic a natural bacterial infection and complement our full suite of functional immune assays which includes a bactericidal assay [23]. Working solutions of LPS were prepared in L-15/5%.

Plated leukocytes were incubated for 24 h with treatments before adding EdU. This incubation time was chosen based on previous studies with a BrdU-based mitogenesis assay in fish [29] and initial kinetics trials in the laboratory (Figure 2) to provide the maximum SI values.

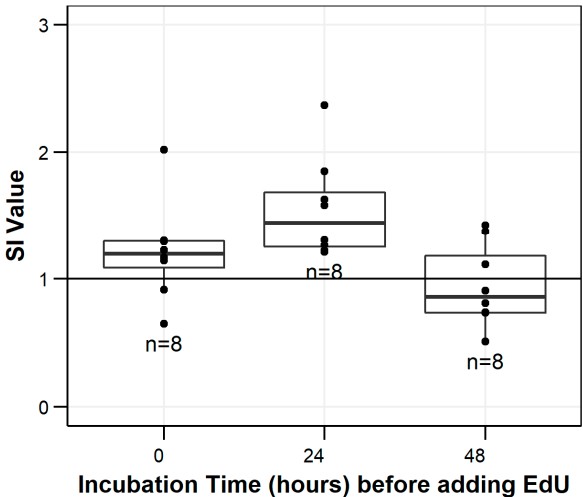

**Figure 2.** Stimulation index (SI) values for different incubation times of leukocytes isolated from laboratory-reared smallmouth bass (*Micropterus dolomieu*). Leukocytes were incubated with LPS and mitogen-free media (negative controls) for 0, 24, and 48 h before adding 5-ethynyl-2'-deoxyuridine (EdU) to detect cell proliferation. The horizontal line indicates the stimulation index (SI) threshold of 1, the dots represent individual data points, and the number indicates sample size (n).

Adult laboratory-reared smallmouth bass was utilized to do initial exposures for determining optimal kinetics of the assay. The smallmouth bass were obtained at age-0 from a fish hatchery in Lake Ariel, PA (Shultz's Fish Hatchery; Lake Ariel, PA, USA) in November 2016 and were housed indoors at the USGS Eastern Ecological Science Center's Leetown Research Laboratory (Kearneysville, WV, USA) in 1287 L circular tanks supplied with flow-through spring water until sampling in February 2018. Fish were reared on disease-free fathead minnows (obtained from Anderson Minnows in Lonoke, AR, USA). Water was heated to 20 °C using a heat exchanger located outside the facility for smallmouth bass while minnows were supplied by water at 13–15 °C. Light was provided by natural and fluorescent light with photoperiod matching natural light cycles. Eight smallmouth bass were sampled to verify kinetics of optimal proliferative response (0, 24, and 48 h incubation with mitogens before adding EdU). Fish were necropsied the same as wild fish above with homogenized anterior kidney tissues being kept at 4 °C overnight before isolating leukocytes the next day.

All incubations occurred at 17 °C in a humidified container (open Zip-loc plastic bag lined with paper towel saturated with deionized water). This temperature was chosen to provide the closest fit to water temperatures from which wild fish were obtained. Following the 24 h incubation with mitogens, 12.5 μL well$^{-1}$ of EdU in unsupplemented L-15 (32 μM) was added to all wells (6.4 μM well$^{-1}$ final concentration) and returned to incubate for 18 h. The mitogenic response was measured after 42 h of incubation with LPS.

### 2.2.4. Detection of Surface IgM

Following the 18 h incubation with mitogens and EdU, leukocytes were treated with an anti-smallmouth bass IgM monoclonal antibody (mAb), previously described [30] to detect cell surface IgM of B lymphocytes. Prior to use, the anti-smallmouth bass IgM mAb

was conjugated (using the Lightning-Link® Rapid Alexa Fluor® 488 kit; Expedeon, San Diego, CA, USA) following the manufacturer's protocol and stored in the dark at 4 °C for up to 18 months. Plated leukocytes were washed, treated with an IgG isotype control (25 µg mL$^{-1}$) and incubated for one hour at room temperature, washed once, and then treated with the AF488-labeled mAb (10 µg mL$^{-1}$) for one hour at room temperature. Cells were washed once before proceeding to cell fixation and the click reaction.

### 2.2.5. Click Reaction

Following cell surface, IgM labeling, incorporation of EdU/cell proliferation was detected by AF647 (using the Click-iT® Plus EdU Flow Cytometry Assay Kit; Molecular Probes, Eugene, OR, USA) with modifications to the manufacturer's protocol (MP 10633). First, 25 µL well$^{-1}$ of saponin-based permeabilization and wash reagent, reaction cocktail, and fixative were used instead of 100 ul well$^{-1}$ as in the manufacturer's protocol. Second, the saponin-based permeabilization and wash reagent was removed from all wells prior to adding the reaction cocktail for the click reaction. The original protocol required adding the reaction cocktail to the saponin wash. Our modification directly exposed the cells to AF647 and increased the signal strength by eliminating the dilution of reaction cocktail created by direct addition to the saponin wash.

Following the click reaction, the reaction cocktail was removed and replaced with saponin-based permeabilization and wash buffer for storage. Plates were stored at 4 °C for up to a week in saponin wash buffer before being analyzed with imaging flow cytometry. The saponin wash buffer was removed and replaced with propidium iodide (PI)/RNase for 15 min to label G2/M mitotic phase cells before running samples through the imaging flow cytometer in the PI/Rnase medium. Intensity of PI fluorescence was used to discriminate cell cycle stage.

### 2.2.6. Imaging and Data Analyses

Following labeling with all fluorochromes, each sample was pooled (two replicate wells per treatment for each fish) and analyzed using an imaging flow cytometer (Amnis FlowSight®; Luminex Corporation, Austin, TX, USA). At least 20,000 events were acquired for each sample. Excitation was set at 100.0 mV for the 642 laser, 5.00 mV for the 785 laser, and 5.00 mV for the 488 laser; fluidics were set to minimum flow speed. Raw image files (.rif) were analyzed using image software with machine learning capabilities (Image Data Exploration and Analysis Software; IDEAS 6.3; Luminex Corporation, Austin, TX). A compensation matrix for AF488, PI/RNase, and AF647 was applied to all .rif files creating compensated image files (.cif) for data analysis.

Primary gating of data analysis files (.daf) isolated individual round cells (R2) that were in focus (R1) based on brightfield images. After initial gating to determine single cells in focus, lymphocytes were selected for analysis based on features of brightfield images (round morphology, low side scatter, nucleus–cytoplasm ratio, nuclear shape) which were applied to the total population using the machine learning module in the imaging software. Proliferating lymphocytes were selected based on concurrent fluorescent signals from PI/RNase (Ch04) indicating they were in the G2/M phase based on intensity of staining and AF647 (Ch11) indicating they were EdU+. The percentage of IgM+/IgM− lymphocytes was determined from the proliferating lymphocyte gate based on AF488 (Ch02) fluorescent positive cells. A template was created containing the primary and secondary gating and was used to analyze all data to provide consistent analytics across samples (Figure 3).

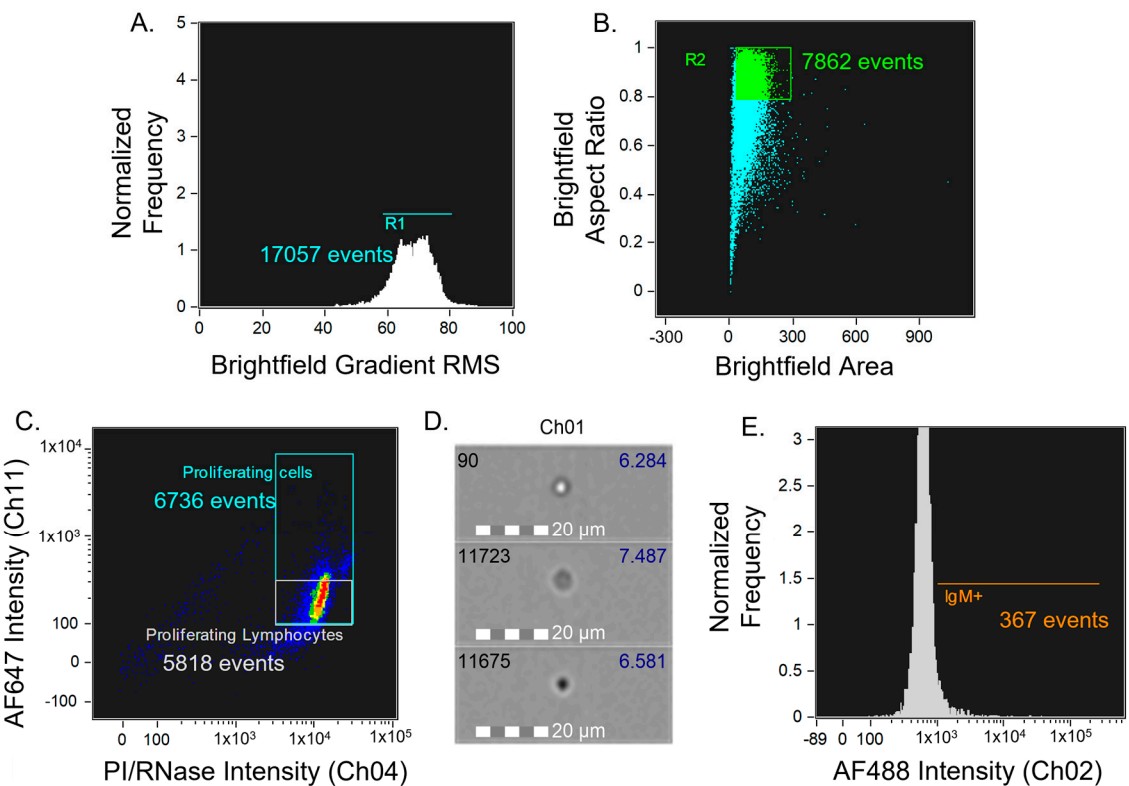

**Figure 3.** Gating strategy for creating batch file analysis template in the image software (IDEAS 6.3). A. Total events/cells (white) in focus were gated as R1 (cyan), B. Out of the R1 population (cyan), single round events/cells were gated as R2 (green), C. Out of the R2 population, proliferating cells (cyan) were gated based on concurrent fluorescent signals from PI/RNase (Ch04) indicating they were in the G2/M phase based on intensity of staining and AF647 (Ch11) indicating they were EdU+, D. Proliferating lymphocytes were selected (gray) for analysis based on features of brightfield images (round morphology, low side scatter, nucleus–cytoplasm ratio, nuclear shape) which were applied to the total population using the machine learning module and distinguished from total proliferating cells (cyan). Cell diameter (μm) is displayed in the top right corner of each image, E. AF488 (Ch02) positives show the number of IgM+ proliferating lymphocytes (orange) out of the proliferating lymphocyte population (gray).

Stimulation index (SI) values were used to compare mitogen-treated with untreated/ negative control cells collected from the same fish. The values represent the fold change in response versus controls; they are a ratio of fluorescent positive cells in treated wells divided by fluorescent positive cells in the negative control wells. Statistics were calculated with various software packages (in R version 3.6.1 [31] using tidyverse [32], ggpubr [33], cowplot [34], reshape [35], gsubfn [36], ggpmisc [37], ggthemes [38], lattice [39], plotrix [40], and FSA [41] packages). This research aimed to analyze differences in immune function among categorical variables such as site, season, and sex and correlations with continuous variables such as age, parasites, and macrophage aggregates to validate its use in future fish health studies. Nonparametric tests were chosen because sample sizes were small (<30), data were not normally distributed, and data could not easily be transformed. Analyses were considered statistically significant when $p \leq 0.05$. Dunn's [42] Kruskal–Wallis test was used for multiple comparisons of sites for each season and *p*-values were adjusted with the Holm method [43,44]. Wilcoxon rank-sum test was used for comparisons among seasons and sex for each site. Spearman rank correlation was used to measure the degree of association between lymphocyte mitogenesis and continuous variables such as age, tissue parasites, and macrophage aggregates. Covariates such as land use were not included in the analysis because there were not enough observations to perform Spearman correlations.

## 3. Results

### 3.1. Mitogenesis Stimulation Indices

When data from all sites and seasons were analyzed, mean SI values varied for both lymphocyte subpopulations (0.71–1.21 for IgM+ and 0.96–1.34 for IgM−. SI varied among sites in the spring for both IgM+ ($p$ = 0.03) and IgM−lymphocytes ($p$ < 0.01; Figure 4A,B). For both lymphocyte subpopulations in the spring, SI was lowest at SB3. The proliferation of IgM+ lymphocytes isolated from CH and SB1 was not significantly different but was higher than in SB3 fish ($p$ = 0.04 and $p$ = 0.05, respectively). IgM− lymphocytes isolated from CH fish responded greater than SB3 fish ($p$ < 0.01) while SB1 was intermediate. There were no significant site differences observed in the fall (Figure 4). Seasonal differences were also observed at CH and SB1 (Figure 4C,D). SI values for IgM+ lymphocytes at CH were greater in spring than in fall ($p$ < 0.01). SI values for IgM− lymphocytes at SB1 were greater in the fall than in spring ($p$ = 0.02).

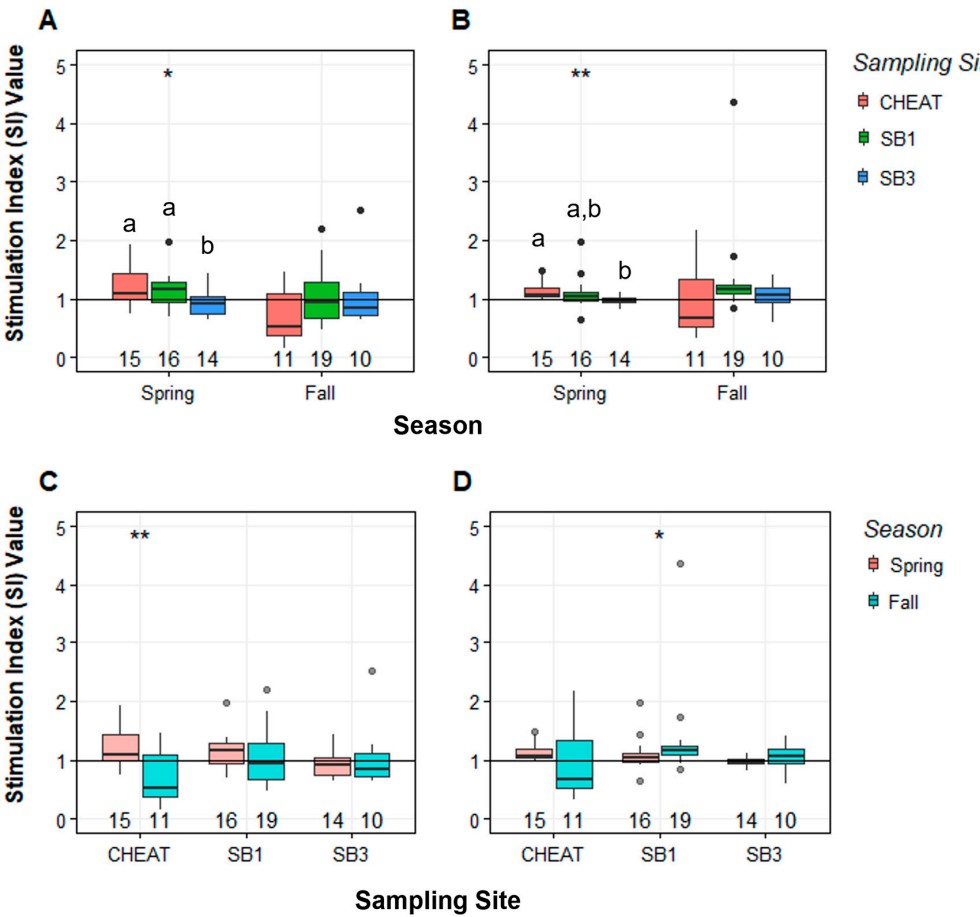

**Figure 4.** Stimulation indices (SI) for IgM+ (left column) and IgM− (right column) lymphocytes isolated from anterior kidney of smallmouth bass (*Micropterus dolomieu*) collected at the Cheat River (CH), South Branch Potomac near Petersburg (SB1), and South Branch Potomac near Moorefield (SB3), West Virginia after exposure to LPS. Site comparisons (**A**,**B**) were made using Dunn's Kruskal–Wallis multiple comparison test, and *p*-values were adjusted with the Holm method. Seasonal comparisons (**C**,**D**) were made using Wilcoxon rank-sum test. Asterisks indicate significant differences between groups (* = $p$ < 0.05, ** = $p$ < 0.01). Superscripts represent significant differences between sites. The horizontal line at 1 indicates the stimulation index (SI) threshold, the dots represent outlying data points (>1.5 times the interquartile range), and the number at 0 indicates sample size (n). Median (line in box) and interquartile range (box) are displayed with the vertical lines indicating highest and lowest values within 1.5 times above and below the interquartile range.

There was a great deal of individual variation in SI for both lymphocyte subpopulations (0.16–2.51 for IgM+ and 0.32–4.36 for IgM−). Some individuals had stimulation indices of at least a two-fold change above background cell proliferation (negative controls), particularly in the fall (Figure 5).

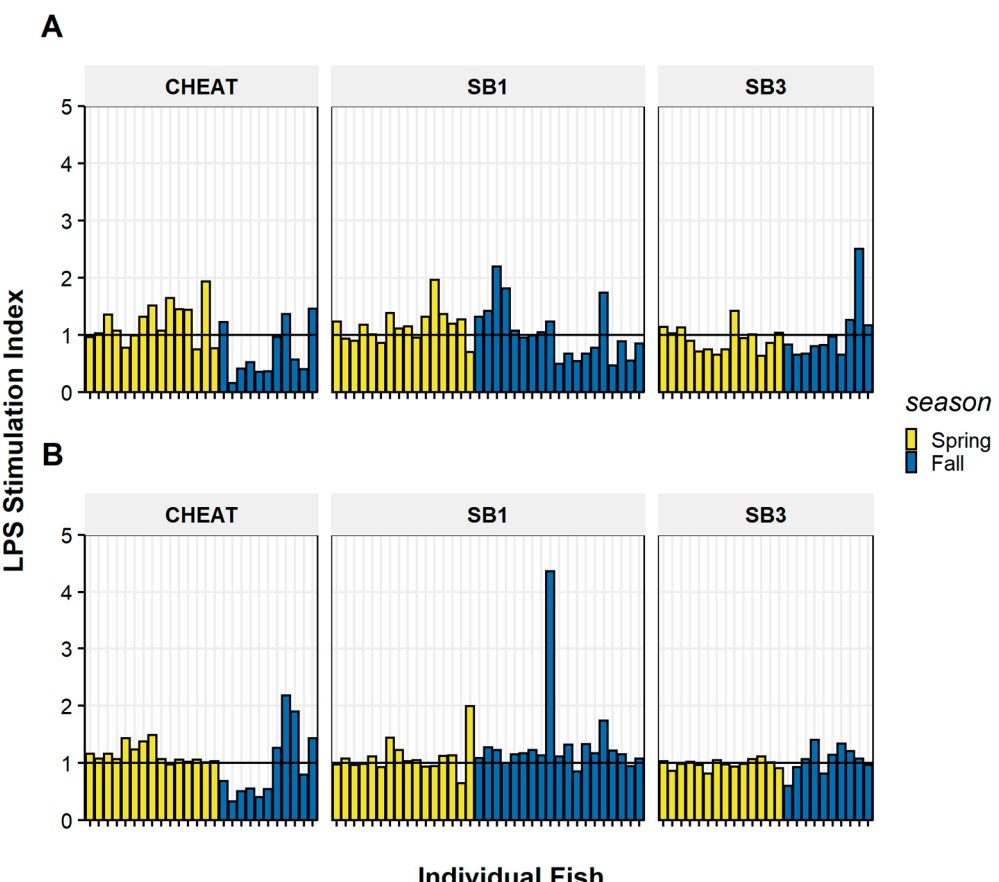

**Figure 5.** Stimulation indices for IgM+ (**A**) and IgM− (**B**) lymphocytes after exposure to LPS for individual smallmouth bass (*Micropterus dolomieu*) during spring (yellow) and fall (blue) for all sampling sites. The horizontal line at 1 indicates the stimulation index threshold.

*3.2. Background Cell Proliferation*

Background proliferation is the replication (intensity of AF647 or EdU incorporation) of anterior kidney lymphocytes when not treated with an immunostimulant (negative controls). It indicates the level of lymphocyte proliferation based on field conditions at the time of collection. Background proliferation varied individually within each site and among sites for both IgM+ and IgM− lymphocytes (Figure 6).

Background proliferation for IgM− lymphocytes was high (>50%) for the majority of fish while low (<5%) for most IgM+ lymphocytes. The percentage of IgM+ background proliferation was higher in spring than fall at all sites ($p < 0.001$) and highest in fish from SB3 ($\bar{x} = 7.04\%$) when compared to the other sites (CH $\bar{x} = 3.17\%$; SB1 $\bar{x} = 2.29\%$; $p \leq 0.05$). There were no differences in IgM+ background proliferation between sites in the fall (SB3 $\bar{x} = 1.20\%$; SB1 $\bar{x} = 0.73\%$; CH $\bar{x} = 1.01\%$). IgM- background proliferation was similar among sites during spring (SB3 $\bar{x} = 57.6\%$; SB1 $\bar{x} = 56.2\%$; CH $\bar{x} = 59.6\%$) but trended higher in the fall at SB3 ($\bar{x} = 67.3\%$) when compared to CH ($\bar{x} = 57.8\%$) and SB1 ($\bar{x} = 54.9\%$); however, this was only significant when compared to SB1 ($p = 0.04$).

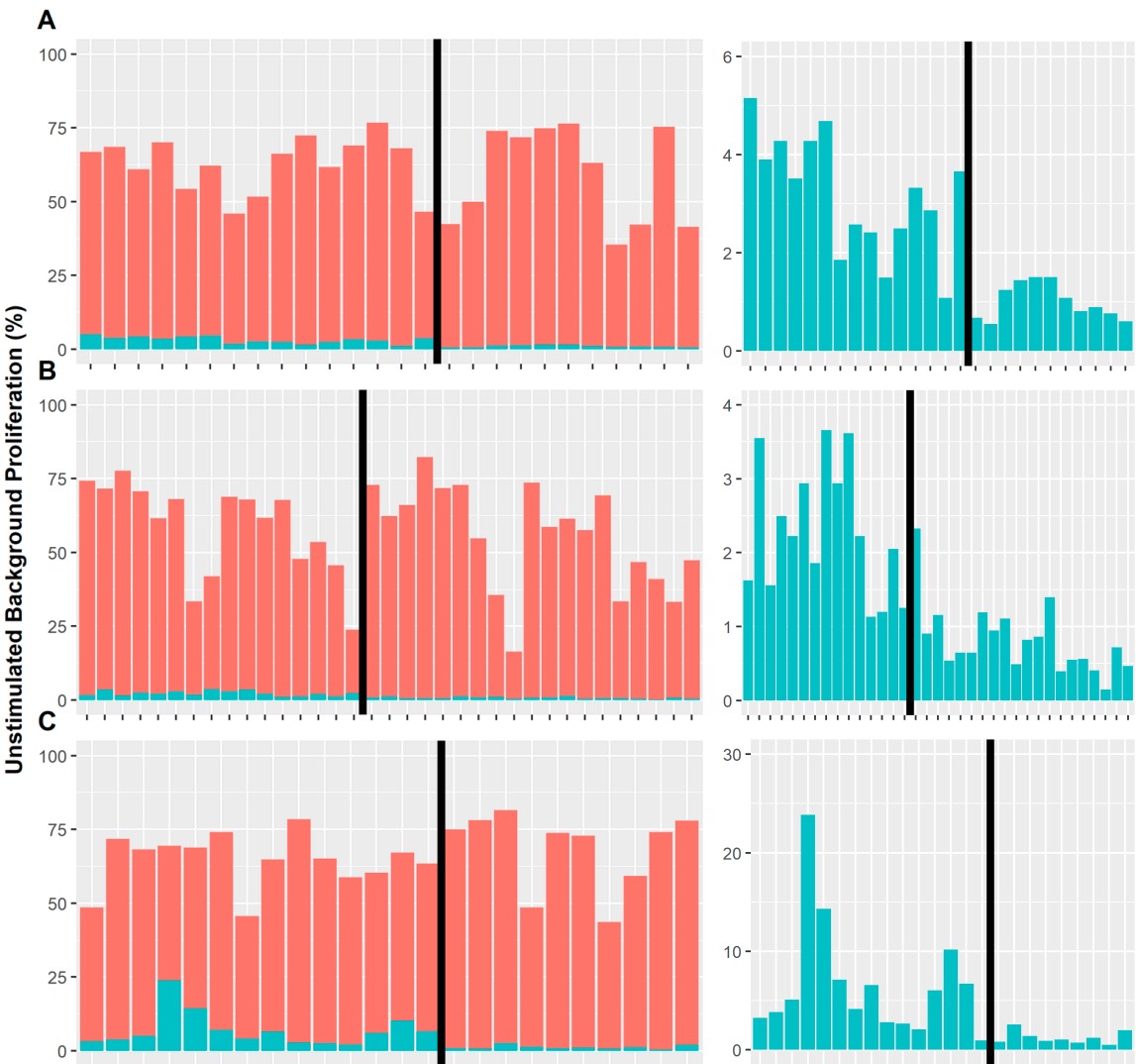

**Figure 6.** Background proliferation responses of IgM− (red) and IgM+ (blue) lymphocytes for individual smallmouth bass (*Micropterus dolomieu*) collected from Cheat River (**A**), South Branch at Petersburg, WV (**B**), and South Branch at Moorefield, WV (**C**). The vertical line separates samples collected in spring (left) and fall (right). The graphs on the right are zoomed in to show the percentages for IgM+ lymphocytes (blue).

In the spring, when all sites were combined there was a negative correlation between unstimulated background cell proliferation and LPS-stimulated mitogenesis responses of IgM+ lymphocytes ($r = -0.41$; $p = 0.005$) and IgM− lymphocytes ($r = -0.61$; $p < 0.0001$). In the fall, this relationship was not significant for IgM+ lymphocytes but was significant ($r = -0.61$; $p < 0.0001$) for the IgM− population. When individual sites were analyzed, there were significant negative correlations during spring and fall for IgM− cells from all sites. For IgM+ lymphocytes, significant correlations were not noted at any site in the fall and only at CH and SB1 in the spring (Table 1). The unstimulated background proliferation suggests activation of lymphocytes was already occurring in the wild fish.

**Table 1.** Correlations between unstimulated background mitogenesis and LPS-stimulated mitogenesis. Spearman correlation method. Only significant results ($p \leq 0.05$) displayed.

| Treatment | Cell Type | Season | Site | r | *p* Value |
|---|---|---|---|---|---|
| LPS | IgM+ | Spring | CH | $-0.68$ | 0.006 |
| LPS | IgM+ | Spring | SB1 | $-0.55$ | 0.026 |
| LPS | IgM$-$ | Spring | CH | $-0.67$ | 0.007 |
| LPS | IgM$-$ | Spring | SB1 | $-0.69$ | 0.003 |
| LPS | IgM$-$ | Spring | SB3 | $-0.64$ | 0.015 |
| LPS | IgM$-$ | Fall | CH | $-0.69$ | 0.019 |
| LPS | IgM$-$ | Fall | SB1 | $-0.58$ | 0.009 |
| LPS | IgM$-$ | Fall | SB3 | $-0.75$ | 0.012 |

### 3.3. Age and Morphometric Characteristics

Lengths and weights were not significantly different among sites, but age did vary (Table 2). Smallmouth bass sampled from CH were significantly older ($\bar{x} = 8.03$) than smallmouth bass collected from either of the South Branch sites (SB1 $\bar{x} = 3.85$; SB3 $\bar{x} = 3.90$; $p < 0.001$). The sex distribution of males to females at the CH site varied with more males (n = 31) than females (n = 8; chi-square test with Bonferroni post hoc correction, $p = 0.018$) being collected. The sex distribution of males to females sampled from the South Branch sites did not differ from expected (1:1).

**Table 2.** Mean ($\pm$standard error) age and morphometric data (n = sample size, mm = millimeter, gm = grams) for adult smallmouth bass (*Micropterus dolomieu*) collected from the Cheat River (CH), South Branch Petersburg (SB1), and South Branch Moorefield (SB3) sites for spring and fall, 2019. Site comparisons made using Dunn's Kruskal–Wallis multiple comparison test and *p*-values adjusted with the Holm method. Superscripts represent significant differences between sites.

| Site | Females (n) | Length (mm) | Weight (gm) | Age (yr) | Males (n) | Length (mm) | Weight (gm) | Age (yr) |
|---|---|---|---|---|---|---|---|---|
| Spring | | | | | | | | |
| CH | 2 | $274 \pm 8$ | $229 \pm 34$ | $7.0 \pm 1.0$ [a] | 18 | $333 \pm 11$ | $477 \pm 51$ | $8.1 \pm 0.4$ [a] |
| SB1 | 7 | $281 \pm 15$ | $317 \pm 53$ | $3.1 \pm 0.4$ [b] | 13 | $309 \pm 13$ | $438 \pm 62$ | $3.4 \pm 0.4$ [b] |
| SB3 | 14 | $318 \pm 17$ | $493 \pm 84$ | $3.9 \pm 0.4$ [b] | 6 | $358 \pm 37$ | $627 \pm 153$ | $3.8 \pm 0.8$ [b] |
| Fall | | | | | | | | |
| CH | 6 | $275 \pm 25$ | $287 \pm 103$ | $8.0 \pm 1.4$ [a] | 13 | $280 \pm 13$ | $283 \pm 51$ | $8.2 \pm 0.7$ [a] |
| SB1 | 11 | $290 \pm 9$ | $316 \pm 30$ | $4.2 \pm 0.3$ [b] | 9 | $289 \pm 11$ | $314 \pm 36$ | $4.7 \pm 0.5$ [b] |
| SB3 | 3 | $318 \pm 81$ | $548 \pm 366$ | $4.7 \pm 2.2$ [a,b] | 8 | $262 \pm 23$ | $308 \pm 98$ | $3.5 \pm 0.7$ [b] |

### 3.4. Age and Mitogenesis

When combining data from all sites, age was not correlated to mitogenesis. The only age-related correlations with mitogenesis were at SB3 for IgM$-$ lymphocytes during spring. Unstimulated background proliferation of IgM$-$ lymphocytes increased with age ($p = 0.017$; Figure 7A) and LPS-stimulated mitogenesis decreased with age ($p = 0.043$; Figure 7B). The data suggest these cells were proliferating prior to mitogen stimulation, and additional cells were not recruited to proliferate. Age was not associated with mitogenesis at the other two sites.

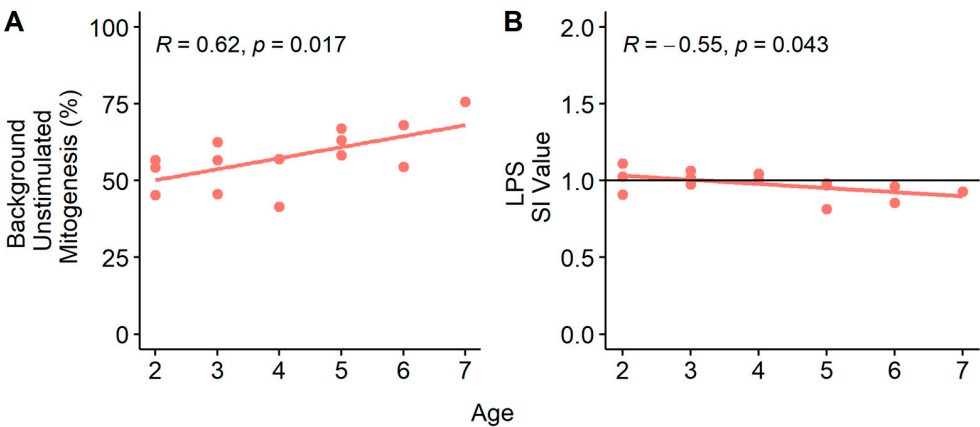

**Figure 7.** Mitogenesis responses of IgM− lymphocytes relative to age. Data were collected in spring 2019 from smallmouth bass (*Micropterus dolomieu*) sampled at a site on the South Branch of the Potomac River in Moorefield, WV (SB3) for (**A**). unstimulated background mitogenesis and (**B**). LPS stimulated mitogenesis. The horizontal line at 1 represents the stimulation index threshold. Data below this line mean proliferation after stimulation did not exceed the negative controls (unstimulated background mitogenesis).

### 3.5. Sex and Mitogenesis

Mitogenesis did not significantly differ among sexes at any of the sites for any season or lymphocyte subpopulation; however, the low sample size could be a factor. At the CH site the median female IgM− response was at or above the SI threshold while the median for males was below the SI threshold, particularly for fall samples (Figure 8A). The same sex differences were not seen for the IgM+ response (Figure 8B). No comparisons were made in the spring since data from only one female was available.

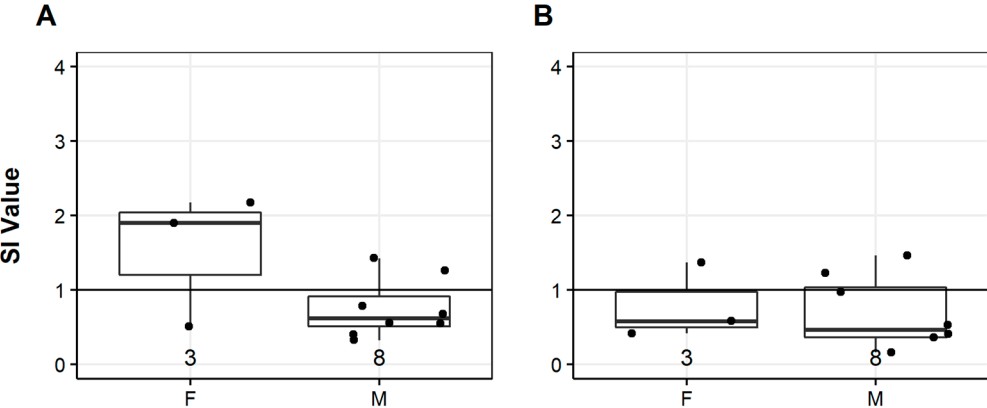

**Figure 8.** Fall 2019 mitogenesis responses of IgM− (**A**) and IgM+ (**B**) lymphocytes in response to LPS stimulation for smallmouth bass (*Micropterus dolomieu*) females (F) and males (M) at the Cheat River site. The horizontal line at 1 indicates the stimulation index (SI) threshold, the dots represent individual data points, and the number indicates sample size (n). Median (line in box) and interquartile range (box) are displayed with the vertical lines indicating highest and lowest values within 1.5 times above and below the interquartile range.

### 3.6. Tissue Parasites and Mitogenesis

The prevalence of spleen parasites, primarily trematodes, varied by the site for spring ($p < 0.01$) and fall ($p = 0.02$). There were no significant differences in parasite prevalence between SB1 and SB3 for spring or fall. Spleen parasites at CH were more prevalent than both SB1 and SB3 in spring ($p = 0.02$ and $p = 0.02$, respectively), and fall ($p = 0.03$ and $p = 0.04$, respectively). There were no significant differences in parasite prevalence between seasons at any of the sites. Significant correlations between the number of parasites in

spleen tissue per square mm and mitogenesis were seen in fish collected from the CH and SB3 sites (Table 3).

**Table 3.** Correlations between tissue parasites in spleen and lymphocyte mitogenesis for smallmouth bass (*Micropterus dolomieu*) collected from the Cheat River (CH) and South Branch Moorefield (SB3) sites. No significant correlations were observed at the South Branch Petersburg (SB1) site.

| Treatment | Tissue Parasites | Cell Type | Season | Site | r | *p* Value |
|---|---|---|---|---|---|---|
| Unstimulated | Spleen | IgM− | Fall | SB3 | −0.68 | 0.030 |
| LPS | Spleen | IgM+ | Spring | CH | −0.51 | 0.054 |
| LPS | Spleen | IgM+ | Fall | CH | 0.62 | 0.041 |

Increased splenic parasites were negatively correlated with unstimulated background response of IgM- lymphocytes at the SB3 site and associated with decreased IgM+ lymphocyte responses in spring and fall at the CH site. No significant correlations with parasites and LPS-stimulated mitogenesis were observed at SB1 or SB3. No significant correlations were found between parasites and unstimulated background proliferation at CH or SB1.

*3.7. Macrophage Aggregates and Mitogenesis*

The prevalence of spleen macrophage aggregates varied by the site for spring ($p < 0.01$) and fall ($p = 0.04$). There were no significant differences in macrophage aggregate prevalence between SB1 and SB3 for either season. Spleen macrophage aggregates at CH were more prevalent than both SB1 and SB3 in the spring ($p = 0.02$ and $p = 0.02$, respectively), and more prevalent than SB3 in the fall ($p = 0.02$). There were no significant differences in macrophage aggregates between seasons at any of the sites. Positive and negative correlations were seen between the prevalence of macrophage aggregates in spleen tissue and LPS-stimulated mitogenesis in the fall and varied based on lymphocyte subpopulation and sampling site (Table 4). Most of the correlations were with IgM− lymphocytes. There were no correlations between mitogenesis and splenic macrophage aggregates for fish from SB1.

**Table 4.** Correlations between splenic macrophage aggregates and lymphocyte mitogenesis for smallmouth bass (*Micropterus dolomieu*) collected from the Cheat River (CH) and South Branch Moorefield site (SB3). No significant correlations were observed at the South Branch Petersburg site (SB1).

| Treatment | Cell Type | Season | Site | r | *p* Value |
|---|---|---|---|---|---|
| Unstimulated | IgM− | Fall | CH | −0.62 | 0.040 |
| LPS | IgM+ | Fall | CH | 0.89 | <0.001 |
| LPS | IgM− | Fall | CH | 0.62 | 0.043 |
| LPS | IgM− | Fall | SB3 | −0.72 | 0.018 |

**4. Discussion**

Results from employing the mitogenesis assay at three West Virginia sites revealed differences in LPS-stimulated and unstimulated background proliferation between sites, seasons, and lymphocyte subpopulations with effects from age, sex, parasites in liver and spleen tissue, and splenic macrophage aggregates (MA). The LPS-stimulated proliferation of smallmouth bass collected in spring showed greater site differences compared to the fall. These differences were likely in part due to differences in physiology (spawning versus recrudescence), climatic factors such as rainfall (i.e., runoff) affecting chemical exposures, factors affecting pathogen and parasite densities, or more likely, a combination of factors. The differences in mitogenesis responses among sites suggest it is important to consider each site for its unique variables affecting immunity because they may differ even within the same river system as revealed with the two South Branch Potomac River sites (SB1 and SB3). Smallmouth bass collected from these sites showed some overlap in their responses but also had noticeable differences. Notably, background proliferation of IgM+ lymphocytes

was significantly higher at SB3. The high background proliferation suggests activation of lymphocytes at SB3 more than SB1, and although fish movement between the two South Branch sites could be a limitation, these sites were chosen to be as independent as possible within a particular system.

Background proliferation for both IgM+ and IgM− lymphocytes in the spring and IgM− lymphocytes in the fall were negatively correlated to LPS-induced proliferation, indicating a relationship exists where increases in background proliferation are associated with less stimulation to mitogens in the laboratory. SB3, our site with the highest background proliferation in spring, also had the lowest stimulated proliferation response in spring. Increases in background proliferation could be an indicator of prior exposure to infectious agents, parasites, or other stressors.

The mitogenic response to LPS varied significantly among individual fish at each site and often was not above the unstimulated background proliferation. This result may suggest lymphocytes were already primed in the field and activated to a level where they could not be stimulated any greater by LPS exposure. The LPS-stimulated SI values in this study may seem low but they are comparable to values for brown bullhead (*Ameiurus nebulosus*; obtained from South Creek in Aurora, North Carolina, USA and acclimated to laboratory conditions in West Virginia, USA before sampling) anterior kidney leukocytes using the same method for mitogen stimulation [45] and brook trout (*Salvelinus fontinalis*; obtained from a hatchery and held in a laboratory in Nova Scotia, Canada before sampling) anterior kidney leukocytes when treated with 200 or 500 μg mL$^{-1}$ LPS [46]. Daly, et al. [46] saw an increase in average SI values to 1.8 but only after adding 5% brook trout plasma and extending incubation time with 200 μg mL$^{-1}$ LPS to 8 days. Average SI values for LPS-stimulated peripheral blood lymphocytes from channel catfish (*Ictalurus punctatus*) held in a laboratory in Texas, USA were higher (2.5–5.4 depending on the cell culture medium supplementation with arginine and/or glutamine); however, they were using peripheral blood lymphocytes and a greater concentration of LPS (500 mg mL$^{-1}$ vs. μg mL$^{-1}$ [47]. It is possible further refinement of the assay could increase the response.

Parasite ratings of liver and spleen tissues and splenic MA ratings were integrated into the analyses to illustrate the application of this new method for detecting immune responses in wild fish. Parasite prevalence was highest at CH compared to the other two sites and the fish from this site were also the oldest. The majority of parasites in the spleen were trematodes. Trematodes have been shown to decrease the proliferative lymphocyte response [48,49] and modulate IgM antibody transcription/production [50–52]. Specifically, trematodes were associated with increased IgM production in roach (*Rutilus rutilus*) from multiple lakes in central Finland [51] and increased transcription of IgM in heart and gills of the Pacific bluefin tuna (*Thunnus orientalis*) sampled from commercial-sized sea cages in Japan [52]. In addition to the high prevalence of trematodes, CH also had the highest LPS-stimulated proliferation of IgM+ lymphocytes which were positively and negatively (depending on the season) correlated with parasite prevalence. The mixed correlations (positive and negative) illustrate the complexities of the immune response and emphasize the need for ancillary data. For instance, depending on the season and likely other factors not measured here such as the stage of infection, the prevalence of intermediate hosts at the site, etc., the increase in trematodes in the field either increased background lymphocyte proliferation and consequently likely decreased the stimulation potential of lymphocytes or primed the IgM+ lymphocytes to respond greater to mitogen stimulation. Host immune response to parasite infection depends on factors such as virulence, type, location, and load, so considering the type of parasite and virulence in future assessments of mitogenesis in wild fish will help interpret results and associations further.

Splenic macrophage aggregates are important to consider in terms of host mitogenesis responses because they can be used as an indicator of exposure to environmental stress [53–55]. They have also been shown to be associated with varied immune responses [56–58]. Deciphering associations between splenic MA and mitogenesis should be easier in future studies when combined with results from the respiratory burst assay.

The respiratory burst assay is part of the full suite of immune function assays adapted for smallmouth bass, which measures reactive oxygen species in phagocytic cells as an indicator of oxidative and environmental stress [23].

It is important to note that the usefulness of in vitro functional immune assays is debated when considered alone to assess immunocompetence due to a lack of standardization and questionable ability to distinguish immunotoxicants from non-immunotoxicants [59,60]. Our results also suggest that using individual functional immune assays, especially in wild fish, may only be valid when considered in context with other collected data such as indicators of disease such as a parasite and macrophage aggregate prevalence in tissues. However, the lymphocyte mitogenesis assay adapted for smallmouth bass was not intended to be used solely as a measure of immunocompetence but to be incorporated into comprehensive fish health assessments where other endpoints were also collected. The purpose is to use the lymphocyte mitogenesis assay as one of many tools or biomarkers for assessing the health of smallmouth bass in the wild and identifying environmental stressors leading to adverse effects. When used in this way, in vitro immune function assays can improve the ability to effectively investigate mechanisms or modes of action through which environmental stressors may modulate the immune response. This assay, together with other immune function assays, and cellular and molecular endpoints could help to identify sublethal effects, identify impacted sites to inform management and direct more in-depth monitoring and assessment.

## 5. Conclusions

The functional mitogenesis assay described in this manuscript uses 5-ethynl-2′-deoxyuridine (EdU) to detect proliferation and imaging flow cytometry with advanced machine learning plus an anti-smallmouth bass mAb to distinguish IgM+ and IgM- lymphocytes. It provides information on the background level of lymphocyte proliferation and the ability of those same lymphocytes to stimulate when exposed to LPS. This method can add valuable information on the adaptive immune status during wild fish health assessments and studies of smallmouth bass. This study is limited to smallmouth bass, but the assay could provide valuable information for other species after optimization. Our analysis emphasizes the importance of including other factors, such as season, site, age, sex, tissue parasite, and macrophage aggregate indicators when designing and planning ecotoxicological studies with immune indicators. A more advanced statistical model with a larger sample size at more sites will be needed to fully understand the potential predictors and covariates of the immune response, but that goes beyond the scope of this paper.

**Author Contributions:** Conceptualization, C.R.S., C.A.O., P.M.M., and V.S.B.; methodology, C.R.S., C.A.O., and V.S.B.; software, C.R.S.; validation, C.R.S. and C.A.O.; formal analysis, C.R.S.; investigation, C.R.S., C.A.O., and H.L.W.; resources, C.R.S., H.L.W., and V.S.B.; data curation, C.R.S.; writing—original draft preparation, C.R.S.; writing—review and editing, C.A.O., H.L.W., P.M.M., and V.S.B.; visualization, C.R.S.; supervision, C.A.O., P.M.M., and V.S.B.; project administration, P.M.M. and V.S.B.; funding acquisition, P.M.M. and V.S.B. All authors have read and agreed to the published version of the manuscript.

**Funding:** This research was funded by the West Virginia Division of Natural Resources (WV DNR) and the U.S. Geological Survey Ecosystem Mission Area's Environmental Health (Contaminant Biology), Cooperative Fish and Wildlife Research Unit, and Species Management (Fisheries) programs.

**Institutional Review Board Statement:** The animal study protocol was conducted in accordance with the US Animal Welfare Act and approved by the Institutional Animal Care and Use Committee of U. S. Geological Survey's Eastern Ecological Science Center (Protocol 2018-003 approved 12 September 2016).

**Data Availability Statement:** The data will be made publicly available through ScienceBase at https://doi.org/10.5066/P9FTUPPX.

**Acknowledgments:** We thank Adam Sperry, Josiah Jensen, and Brenna Raines for their assistance with fish necropsies; WV DNR biologists Brandon Keplinger, Jim Walker, Dustin Smith, and their crews for collecting fish; Stephanie Gordon (USGS) for the production of maps and land cover information; and Clay Raines (USGS) for statistical advice. Any use of trade, firm, or product names is for descriptive purposes only and does not imply endorsement by the U.S. Government.

**Conflicts of Interest:** The authors declare no conflict of interest. The funders had no role in the design of the study; in the collection, analyses, or interpretation of data; in the writing of the manuscript; or in the decision to publish the results.

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
