# Peer review of "Application of a Lipopolysaccharide (LPS)-Stimulated Mitogenesis Assay in Smallmouth Bass (Micropterus dolomieu) to Augment Wild Fish Health Studies"

_fishes, doi:10.3390/fishes8030159_

Round 1

Reviewer 1 Report

This manuscript provides a nice addition to the toolkit for immunological studies in fish.  The methods presented will allow for broader studies of the impact of various biotic and abiotic factors on fish immunity, specifically in wild bass populations.  In general the methods provides sufficient details, results and figures were clear, and writing was easy to follow.  I had only a few minor suggestions:

1) The introduction is a bit broad and lacking in focus.  The first couple of paragraphs are also have some redundancies.

2) Details are lacking in the statistical analysis section.  What were the terms in the linear model?  What error distribution was used?  Also, R2 has previously been shown to provide spurious results for model selection.  Other approaches such as AIC are recommended.

3) The results presented a variety of separate analysis.  There are some concerns with inflated type I error due to multiple tests.  Also, section 3.8 provides a nice synthesis of what predictors are important , and it seems like this should drive the focus of the other results sections.

4) Figure 3 seems to be the output of flow-cytometry software.  As a result axis labels are unclear.  It is recommended these be replaced with more informative labels.   

Reviewer 2 Report

see attachment

Reviewer 3 Report

Reviewer’s comments

Manuscript Number:

Title: Application of an LPS-stimulated mitogenesis assay in smallmouth bass (Micropterus dolomieu) to augment wild fish health studies

Journal: Fishes

Summary

In this article, the authors describe and test a functional mitogenesis assay using 5-ethynl-2’-deoxyuridine (EdU) to detect and measure adaptive immunity in wild smallmouth bass (Micropterus dolomieu). The associations of lymphocyte mitogenesis with the season, sex, age, tissue parasites, and macrophage aggregates were also evaluated using multiple linear regression. Wild smallmouth bass samples were collected from three sites in West Virginia and the land cover data was downloaded from the National Land Cover Database. After the fish samples were collected, histological analysis and leucocyte isolation were performed. Regarding the mitogenesis analysis, the authors used an EdU-bases assay protocol optimized for smallmouth bass. The manuscript is interesting and contains useful information. Regarding the results, the author stated that 1) the mitogenic response to LPS varied significantly among individual fish at each site, 2) site is a significant factor when considering using in vitro assays to measure immune function, and 3) season, sex, and weight were the most significant predictors of the unstimulated lymphocyte response for all sites combined. However, some improvements should be made before accepting this paper. I suggested this work may be “minor revision” for publication in “Fishes”. Specific comments and general comments are given below:

General comments

1.      Page 1, Abstract. The abstract of the manuscript should be rewritten. The authors should state briefly the purpose of the research, the principal results, and major conclusions.

2.      Page 1-2, Introduction. This section is too long. The authors are suggested to remove some of the background material. Cite several comprehensive review articles in this area, get to the problem or knowledge gap sooner, and tell the reader how your study filled the gap.

3.      Page 3, Materials and Methods. This section was well written and easy to follow. However, the authors are suggested to provide reasons for:

-        sampling site selection.

-        the use of multiple linear regression (MLR) for the association of lymphocyte mitogenesis with predictive variables. The authors are suggested to check the assumptions for MLR. Otherwise, nonlinear regression analysis may be appropriate for this study. 

4.      Page 8, Results. As the data were not normally distributed, the authors are suggested to present the median along with the percentiles of the data. In addition, the statistical inference results from the small sample size and the use of MLR for non-normally distributed data are rather difficult to determine if a particular outcome is a true finding. Therefore, the authors need to provide strong justification for this or consider other methods, such as nonlinear regression analysis or data transformation.

5.      Page 13, Discussion. The authors are suggested to provide more discussion based on their findings. Please see the specific comment below.

6.      Authors are advised to state the limitations of this study. In addition, providing a conclusion in a separate section is highly recommended.

Specific comments

1.     Line 79-83, The authors are suggested to revise this sentence as it is not very clear to the reader.

2.     Table 1, please provide the median value of the data.

3.     Figure 7, please provide detailed information about the boxplot. Does the middle line represent the mean or the median?

4.     Line 472-473, “With all sites combined, season, sex, and weight were the most significant predictors of the unstimulated lymphocyte response…” If these findings are true, then further discussion of how these factors affect lymphocyte response will help the reader to understand the significance of these findings. If possible, the author can also provide some examples.

5.     Line 465-469 that “Each site had distinct factors that were important when considering the unstimulated background proliferation or the baseline lymphocyte response from fish at each site. At Cheat, season, weight, and total parasites were significant predictors of unstimulated background proliferation.” Discussion on the site characteristics may provide more insights into this phenomenon.

6.     Line 483-498, “Certain parasites have been shown to decrease the proliferative lymphocyte response …” Instead of just citing the previous work, the authors could help readers to understand how the immune response to parasite infection depends on factors by providing some examples.

7.     Line 518-530, “More work, including in vitro exposures, gene expression analysis of immune- and contaminant-related genes, and analysis of surface water and tissue chemical concentrations, is being done to…” This research looks like unfinished research. The authors may consider removing this paragraph or finishing their work first prior to publishing it.

Round 2

Reviewer 2 Report

thank you for addressing the initial review. I have no further comments.